# DEEP ACCURATE SOLVER FOR THE GEODESIC PROBLEM

## ABSTRACT

A high order accurate deep learning method for computing geodesic distances on surfaces is introduced. We consider two main components for computing distances on surfaces; A numerical solver that locally approximates the distance function and an efficient causal ordering scheme by which surface points are updated. The proposed method exploits a dynamic programming principle which lends itself to a scheme with quasi-linear computational complexity. The quality of the distance approximation is determined by the local solver and is the main focus of this paper. A common approach to compute distances on continuous surfaces is by considering a discretized polygonal mesh approximating the surface, and estimating distances on the polygon. With such an approximation, the exact geodesic distances restricted to the polygon are at most second order accurate with respect to the distances on the corresponding continuous surface. Here, by *order of accuracy* we refer to the rate of convergence as a function of the average distance between sampled points. To improve the accuracy, we consider a neural network based local solver which implicitly approximates the structure of the continuous surface. The proposed solver circumvents the polyhedral representation, by directly consuming sampled mesh vertices for approximation of distances on the sampled continuous surfaces. We supply numerical evidence that the proposed learned update scheme, with appropriate local numerical support, provides better accuracy compared to the best possible polyhedral approximations and previous learning based methods. We introduce a trained solver which is third order accurate, with quasi-linear complexity in the number of sampled points.

## 1 INTRODUCTION

Geodesic distance is defined as the length of the shortest path connecting two points on a surface. It can be considered as a generalization of the Euclidean distance to curved manifolds. The approximation of geodesic distances is used as a building block in many applications. It can be found in robot navigation (Kimmel et al., 1998; Kimmel & Sethian, 2001), and shape matching (Ion et al., 2008; Elad & Kimmel, 2001; Shamai & Kimmel, 2017; Panozzo et al., 2013), to name just a few examples. Thus, for effective and reliable use, computation of geodesics is expected to be both *fast* and *accurate*.

Over the years, many methods have been proposed for computing distances on polygonal meshes that compromise between the accuracy of the distance approximation and the complexity of the algorithm. One family of algorithms for computing distances in this domain is based on solutions to the *exact* discrete geodesic problem introduced by Mitchell et al. (1987). This problem is defined as that of finding the exact distances on a polyhedral mesh. The algorithms introduced so far for solving the discrete geodesic problem involve substantially higher than linear complexity which makes them impractical for operating on surfaces sampled by a large number of vertices. At the other end, a popular family of methods for efficient approximation of distances known as *fast marching*, involves quasi-linear computational complexity. These methods are based on the solution of the *eikonal equation* and consists of two main components, a heap sorting strategy and a local causal numerical solver, often referred to as a numerical update procedure. Fast marching, originally introduced for regularly sampled grids (Sethian, 1996; Tsitsiklis, 1995), was extended to triangulated surfaces in Kimmel & Sethian (1998). While operating on curved surfaces approximated by triangulated mesh, the first proximity neighbors of a vertex in the mesh are used to locally approximate the solution of

an eikonal equation, resulting in a *first-order-accurate scheme* in terms of a typical triangle's edge length denoted as $h$.

It can be easily shown that the exact geodesic distances computed on a polygonal mesh approximating a continuous surface would be at most a second order approximation of the corresponding distances on the surface (see Appendix A.1). To overcome the second order limitation, we extend the numerical support about each vertex beyond the classical one ring approximation, and utilize the universal approximation properties of neural networks. We develop a neural network based local solver that overcomes the second order approximation limitation induced by geodesics restricted to polyhedral meshes. We exploit the low complexity of the well-known dynamic programming update scheme (Dijkstra, 1959), and combine it with a novel neural network-based solver, resulting in an *efficient* and *accurate* method.

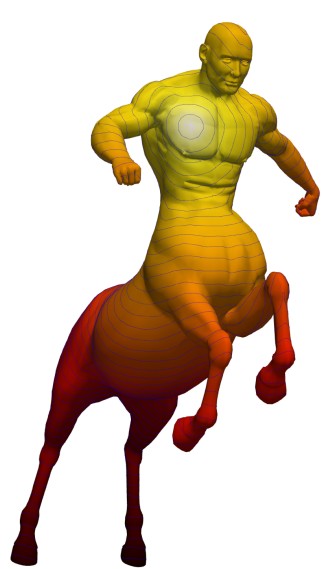

In a related effort (Lichtenstein et al., 2019), a neural network based local solver for the computation of geodesic distances was proposed. We improve upon Lichtenstein's $\mathcal{O}(h^2)$ approach by extending the local neighborhood numerical support, and refining the network's architecture to obtain $\mathcal{O}(h^3)$ accuracy at similar linear complexity.

The proposed local solver is trained in a supervised manner using ground truth examples. Since geodesics can not be derived analytically except for a limited set of surfaces like spheres and planes, we propose a multi-hierarchy ground truth generation technique. The suggested framework uses distance approximations on high resolution sampled meshes to better approximate distances on lower resolution meshes. We thereby utilize our ability to compute high order solvers to generate accurate training examples at low resolution.

Figure 1: Geodesic distance from a single source point on a surface. Our method produces highly accurate distance maps while operating in quasi-linear time.

## 1.1 CONTRIBUTIONS

We develop a *fast* and *accurate* geodesic distance approximation method on surfaces.

- For fast computation, we use a distance update scheme (Algorithm 1) that guarantees quasi-linear computational complexity.
- For accurate approximation, we develop a neural network based local solver with a wide local neighborhood support that operates directly on the sampled mesh vertices.
- To provide accurate ground truth distances required for training our solver, we propose a novel data generation bootstrapping procedure.

## 2 RELATED EFFORTS

Given a domain $\Omega \subset \mathbb{R}^n$ and a curve $\Gamma \in \Omega$, the predominant approach for generating distance functions from the curve $\Gamma$ to all other points in $\Omega$, is to find a function $\phi : \Omega \rightarrow \mathbb{R}$ which satisfies the *eikonal equation*,

$$\begin{aligned} |\nabla\phi(x)| &= 1, & x \in \Omega \setminus \Gamma \\ \phi(x) &= 0, & x \in \Gamma. \end{aligned} \tag{1}$$

Due to the non-linearity and hyperbolicity of this partial differential equation (PDE), solving it directly is a challenge. Common solvers sample the continuous domain and approximate the solution on the corresponding discretized domain while being consistent with *viscosity* solutions.

**Fast Eikonal Solvers**. In Sethian (1996); Tsitsiklis (1995), quasi-linear algorithms for approximating distances on regularly sampled grids were introduced. These algorithms involve $\mathcal{O}(N \log(N))$ complexity, where $N$ is the number of points on the grid. For example, the *fast marching algorithm*

consists of two main parts, a numerical solver that locally estimates the distance function, by approximating a solution of an eikonal equation, and an ordering scheme that determines which points are visited at each iteration. Over the years, more sophisticated local solvers have been developed which utilize wide local support and lead to second (Sethian, 1999) and third (Ahmed et al., 2011) order accurate methods. Kimmel & Sethian (1998) extended the fast-marching scheme to approximate first-order accurate geodesic distances on triangulated surfaces. Another prominent class of numerical solvers is the fast sweeping methods (Kimmel & Maurer, 2006; Zhao, 2005; Li et al., 2008; Weber et al., 2008), iterative schemes that use alternating sweeping ordering. These methods have an asymptotic complexity of $\mathcal{O}(N^2)$, thus, they are usually considerably slower since they require many sweeps to converge to accurate results in worst case scenarios (Hysing & Turek, 2005).

**Geodesics in Heat**. Instead of directly solving the hyperbolic eikonal equation, the heat method presented in Crane et al. (2013) solves a linear elliptic PDE for heat transport. The suggested approach requires the solution of a diffusion equation and a Poisson equation. An advantage promoted in that paper, is that the required linear systems can be pre-factored per mesh, which leads to computational efficiency when calculating distance maps from multiple sources. It has been empirically shown that the heat method leads to first order approximation of geodesic distances on average. The distance functions produced with the heat method are smoother than those generated with fast marching and are less accurate near the discontinuities of the distance functions.

**Window Propagation Methods**. Trying to solve the discrete geodesic problem, Mitchell et al. (1987) proposed a $\mathcal{O}(N^2 \log(N))$ complexity algorithm known as MMP. This algorithm was the first method introduced for computing exact distances on non-convex triangulated surfaces. In its original formulation, it calculates the distance from a single source to all other points on the polygonal mesh. The main idea of this algorithm is to track groups of shortest paths that can be atomically parameterized. This is achieved by dividing each mesh edge into a set of intervals, which are referred to as *windows*. This quadratic algorithm is computationally demanding and challenging to code. In fact, the first implementation was introduced 18 years after its publication, by Surazhsky et al. (2005). Over the years, many improvements to the exact geodesic scheme were suggested (Xu et al., 2015; Ying et al., 2014; Trettner et al., 2021). For example, the Vertex-oriented Triangle Propagation (VTP) algorithm (Qin et al., 2016), which by sorting out superfluous windows is considered to be one of the fastest exact geodesics algorithms on polyhedral surfaces.

**Deep Learning Based Methods**. A number of recent papers exploited neural networks approximation capabilities to numerically solve PDEs (Greenfeld et al., 2019; Hsieh et al., 2019; bin Waheed et al., 2021). Similar to our strategy, Lichtenstein et al. (2019) proposed a deep learning based method for geodesic distance approximation. It uses a heap sort ordering scheme while introducing a neural network based local solver. For each distance evaluation of a target point $p$, a local neighborhood is obtained as input to the solver. This neighborhood consists of all vertices connected to $p$ by a path with at most 2 edges, which we refer to as second-ring neighborhood. The proposed method showed second-order accuracy, similar to the exact geodesic method (Mitchell et al., 1987), while operating at quasi-linear $\mathcal{O}(N \log(N))$ computational complexity.

**Curve Shortening Methods**. A geodesic is defined as the path connecting two points on a surface, which is characterized by having zero geodesic curvature at each point along the curve. Between two points on a surface, there can be multiple geodesic paths with different corresponding distances. The geodesic distance is defined as the length of the minimal geodesic connecting each point on the surface to some source points at which the distance is defined to be zero. A geodesic path can be extracted by starting with an arbitrary path on the surface and applying a length shortening method. Kimmel & Sapiro (1995) presented a curve shortening flow that fixes the two endpoints of the curve at each iteration and minimize the geodesic curvature. Sharp & Crane (2020) presented a curve shortening method for triangulated meshes by considering paths constrained to the mesh edges and applying intrinsic edge flips. These local refinement methods converge an initial guess into a geodesic which is not necessarily the minimal one.

## 3 GEODESIC DISTANCES: $\mathcal{O}(h^3)$ ACCURATE AT QUASI-LINEAR COMPLEXITY

We present a neural network based method for approximating accurate geodesic distances on surfaces. Similar to most dynamic programming methods, like the fast marching scheme, the proposed method consists of a numerical solver that locally approximates the distance function $u$, and an or-

dering scheme that defines the order of the visited points. Here, the points are divided into three disjoint sets.

1. **Visited**: points where the distance function $u(p)$ has already been computed and will not be changed.

2. **Wavefront**: points where the computation of $u(p)$ is in progress and is not yet fixed.

3. **Unvisited**: points where $u(p)$ has not yet been computed.

---

**Algorithm 1** Distance Updating Scheme

---

1: **Definitions:**
    $S$ - Set of all source points
    $p$ - point on the surface
    $u(p)$ - minimal distance from sources to $p$
2: **Initialize:**
    $u(p) = 0$, tag $p$ as *Visited*; $\forall p \in S$
    $u(p) = \infty$, tag $p$ as *Unvisited*; $\forall p \notin S$
    Tag all *Unvisited* points adjacent to *Visited* points as *Wavefront*
3: **repeat**
4:     **for** $p \in$ *Wavefront* **do**
5:         Approximate $u(p)$ based on *Visited* points
6:         Tag $p$ as *Wavefront*
7:     **end for**
8:     Tag the least distant *Wavefront* point $p'$ as *Visited*
9:     Tag all *Unvisited* neighbors of $p'$ as *Wavefront*
10: **until** all points are *Visited*.
11: **Return** $u$

---

The distances at the sampled surface points are computed according to Algorithm 1, where Step 5 of the scheme is performed by the proposed local solver. When applied to a target point $p \in$ *Wavefront*, the local solver uses a predefined maximum number of *Visited* points. These *Visited* points are chosen from the local neighborhood and are not related to the number of points on the mesh; hence, a single operation of our solver has constant complexity. Since, within our dynamic programming setting, the proposed method retains the heap sort ordering scheme, the overall computational complexity is $\mathcal{O}(N \log(N))$.

Section 3.1 introduces the operation of the local solver, presents the required pre-processing and elaborates on the implementation of the neural network. Section 3.2 explains how the dataset is generated and the network weights are optimized. Section 3.3 details how ground truth distances are calculated when no analytic closed form is available.

### 3.1 LOCAL SOLVER

We present an application of a novel local neural network-based solver. It can be used as a distance approximation method in Step 5 of Algorithm 1. When the solver is applied to a given point $p \in$ *Wavefront*, it receives as input the coordinates and distance function values of its neighboring points. The neighboring points, denoted by $\mathcal{N}(p) = \{p_1, p_2, ..., p_M\}$, are defined by all vertices connected to $p$ by a path of at most 3 edges, which is often referred to as third ring neighborhood. Based on the information from the *Visited* points in $\mathcal{N}(p)$, the local solver approximates the distance function $u(p)$. This way, we keep utilizing the order of updates that characterizes the construction of distance functions.

As mentioned earlier, for a given target point $p$ and neighboring points $\{p_i\}_{i=1}^{M} \subset$ *Visited* $\cap \mathcal{N}(p)$, the input to our solver is $\{(x_{p_i}, y_{p_i}, z_{p_i}, u(p_i))\}_{i=1}^{M} \cup \{(x_p, y_p, z_p)\}$. To address the solver's generalization capability and to handle diverse possible inputs, we transform the input to the neural network into a canonical representation. To this end, we design a preprocessing pipeline. The coordinates are centered with respect to the target point, resulting in relative coordinates $(x_{p_i} - x_p, y_{p_i} - y_p, z_{p_i} - z_p)$, and $\min_j \{u(p_j)\}$ is subtracted from the values of the distance function $\{u(p_i)\}_{i=1}^{M}$. After the input is centered, it is scaled so that the mean L2 norm of the coordinates is of unit size. Last, a $SO(3)$

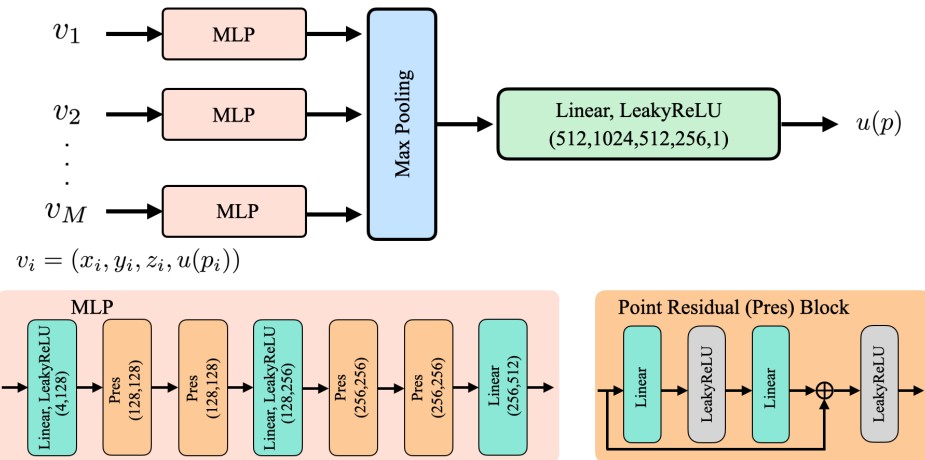

Figure 2: The proposed network architecture described in Section 3.1. The input coordinates and distance function are in their canonical form, after translation rotation and scale.

rotation matrix is applied to the coordinates so that their first moment is aligned with a predefined direction. The processed input is fed into the neural network and the output is further processed to reverse the centering and scaling transformations.

Our input neighborhood has no fixed order and can be viewed as a set. To properly handle our unstructured set of points, we prefer our neural network output to be permutation invariant. Our architecture, shown schematically in Figure 2, is based on the Pointnet model (Qi et al., 2017) and consists of three main components. (1) A shared weight encoder that lifts the 4-dimensional input to 512 features using residual multi-layer perceptron (MLP) blocks (Ma et al., 2022), (2) a per-feature max pooling operation that results in a single 512 feature vector, and (3) a fully connected regression network of dimensions $(512, 1024, 512, 256, 1)$ that outputs the desired target distance.

## 3.2 TRAINING THE LOCAL SOLVER

We use a customary supervised training procedure, using examples with corresponding ground-truth distances. These ground truth distances are obtained by applying a bi-level sampling strategy, as detailed in Section 3.3. Given an input $\{(x_{p_i}, y_{p_i}, z_{p_i}, u(p_i))\}_{i=1}^M$, our network is trained to minimize the difference between its output and its corresponding ground truth, denoted by $u_{gt}(p)$. To develop a reliable and robust solver, we create a diverse dataset that simulates a variety of scenarios. We construct this dataset by selecting various source points and sampling local neighborhoods at different random positions relative to the sources. According to the causal nature of our algorithm, we build our training examples, such that a neighboring point $p'$ is defined as *Visited* and is allowed to participate in the prediction of $u(p)$ if $u_{gt}(p') < u_{gt}(p)$. The network's parameters $\Theta$ are optimized to minimize the Mean Square Error (MSE) loss

$$L(\Theta) = \frac{1}{K} \sum_{j=1}^K \left( f_\Theta(\{(x_{p_{i,j}}, y_{p_{i,j}}, z_{p_{i,j}}, u(p_{i,j}))\}_{i=1}^M) - u_{gt}(p_j) \right)^2, \tag{2}$$

where $K$ is the number of examples in the training set and $p_{i,j}$ corresponds to the $i^{th}$ neighbor of the target point $p_j$. The coordinates and distances used in our training procedure are in their canonical form, after being translated, rotated and scaled, as explained in Section 3.1.

## 3.3 LEARNING TO AUGMENT

Exact distances on continuous surfaces are given by analytic expressions for a very limited set of continuous surfaces; namely, for spheres and planes. Since our solver is trained on examples containing ground truth distances, an additional approximation algorithm must be considered to generate our training examples for general surfaces. Currently, the most accurate axiomatic method for

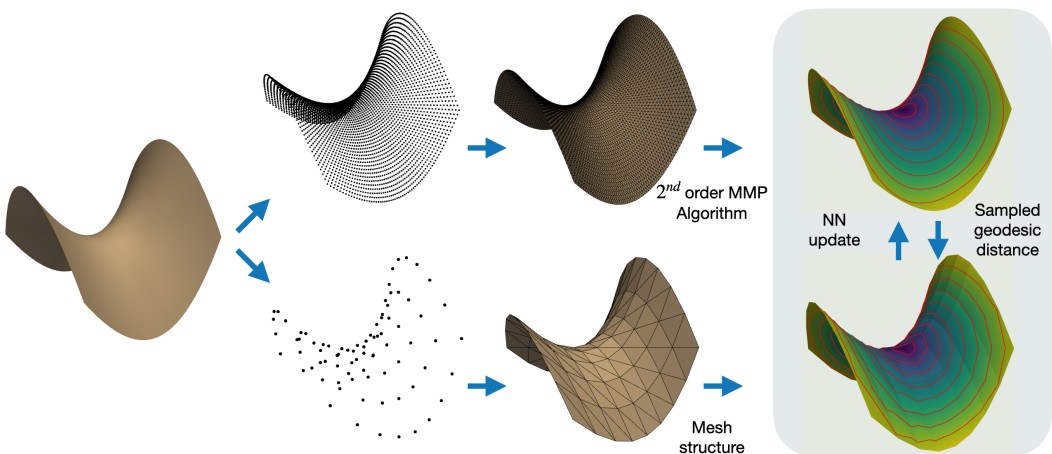

Figure 3: Bootstrapping by training. Distance values computed for a high $h^2$-resolution sampled mesh of a continuous surface with an $r$ accurate scheme yields $\mathcal{O}(h^{2r})$ accurate distances given at the mesh points. The mesh can then be sampled into a lower $h$-resolution mesh of the same continuous surface, while keeping the corresponding $\mathcal{O}(h^{2r})$ accurate distances at the vertices. See text for an elaborated discussion regarding data augmentation at high resolution and training more accurate update procedures at the low resolution.

distance computation is the MMP algorithm, which computes "exact" polyhedral distances. Considering polyhedral surfaces as sampled continuous ones, the "exact" distances on triangulated surfaces are $2^{nd}$ order accurate with respect to the edge length $h$. Therefore, the MMP is an $\mathcal{O}(h^2)$ accurate method. In order to train our network with more accurate than $\mathcal{O}(h^2)$ distances for general smooth surfaces, we resort to the following bootstrapping idea.

We introduce a multi-resolution ground truth boosting generation technique that allows us to obtain ground truth distances of any desired order. The underlying idea is that distances computed on a mesh obtained from a denser sampling of the surface are a better approximation to the distances on the continuous surface.

When generating examples from a given surface, two sampling resolutions of the surface are obtained and corresponding meshes are formed, denoted by $S_{dense}$ and $S_{sparse}$, respectively. Distances are computed on the high-resolution mesh $S_{dense}$ and the obtained distance map is sampled at $S_{sparse}$.

Consider $h_{dense}, h_{sparse}$ that correspond to the mean edge length of the polygons $S_{dense}, S_{sparse}$, so that,

$$h_{dense} = h_{sparse}^q . \qquad (3)$$

The distances computed by an approximation method of order $r$ on $S_{dense}$ are $r$ order accurate $\mathcal{O}(h_{dense}^r)$. Therefore, the same approximated distances, sampled at the corresponding vertices of $S_{sparse}$, have $\mathcal{O}(h_{sparse}^{qr})$ accuracy.

Using the distance samples of the polyhedral distances obtained by the MMP algorithm while requiring $q \geq 2$ in Equation (3), allows us to generate distance maps that are at least fourth-order accurate. By considering these approximated distances as our ground truth, training examples can be generated from $S_{sparse}$ as described in Section 3.2 which allow us to properly train a third-order accurate method. The iterative application of this process allows us to generate accurate ground truth distances to properly train solvers of arbitrary order. For example, after training a $3^{rd}$ order solver, we can apply the same process while replacing the MMP with our new solver to generate a $\mathcal{O}(h^6)$ ground truth distances and train a solver up to $6^{th}$ order. For a schematic representation of this technique, see Figure 3.

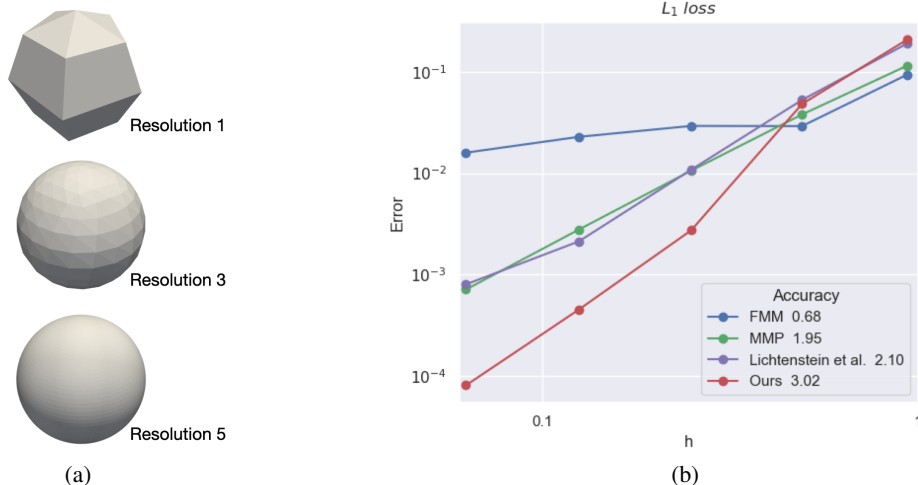

Figure 4: Order of accuracy: (a) Mesh approximations of a unit sphere with different edge resolutions. (b) Plots showing the effect of the edge size on the errors. The accuracy of each scheme is associated with its corresponding slope.

## 4 NUMERICAL EVALUATION

### 4.1 ANALYTICALLY COMPUTED GROUND TRUTH

Geodesic distances on spheres can be calculated analytically. Therefore, they are well suited for the evaluation of our method. For two given points $a = (x_a, y_a, z_a), b = (x_b, y_b, z_b)$ lying on a sphere of radius $r$, the geodesic distance between them is defined by

$$u(a,b) = r \arccos\left(\frac{x_a x_b + y_a y_b + z_a z_b}{r^2}\right). \quad (4)$$

To train our solver, we first randomly sampled spheres at different resolutions and obtained a triangulated version of them. Using the distances calculated by Equation (4), we then created a rich data set and applied a training procedure as presented in Section 3.2. To evaluate our method, we create a hierarchy of sphere resolutions as illustrated in Figure 4a.

As described in Osher & Sethian (1988), we assume that the exact solution $u(a, b)$ can be written as

$$u(a,b) = u_h(a,b) + Ch^R + \mathcal{O}(h^{R+1}), \quad (5)$$

where $C$ is a constant, $u_h$ is defined as the approximate solution on a mesh with a corresponding mean edge length of $h$, and $R$ is the order of accuracy. For two given mesh resolutions of the same continuous surface $S_1, S_2$ with corresponding $h_1, h_2$, we can estimate our method's order of accuracy by

$$R = \log_{\frac{h_1}{h_2}}\left(\frac{u - u_{h_1}}{u - u_{h_2}}\right). \quad (6)$$

The evaluation of our method is shown in Figure 4b, where the slope of the line indicates the order of accuracy $R$. It can be seen that our method has a higher order of accuracy than the classical fast marching, the exact geodesic method, and the previous deep learning method proposed by Lichtenstein et al.

### 4.2 GENERALIZATION TO POLYNOMIAL SURFACES

We evaluated our method on second order polynomial surfaces. In general, there is no closed form analytical expression for geodesic distances on these surfaces. To train our solver, we generated a wide variety of polynomial surfaces and obtained an accurate approximation of their geodesics for a range of sampling resolutions, as described in Section 3.3. After obtaining an accurate geodesic distance map, we created a training set and trained our model according to Section 3.2. An evaluation of our method on surfaces from this family is shown in Table 1 and Figure 5.

Table 1: Polynomial surfaces: Quantitative evaluation conducted on $2^{nd}$ order polynomial surfaces. The errors were computed relative to the polyhedral distance projected from high-resolution sampled meshes, as described in Section 3.3.

| Surface | $L_1$ | | | | $L_\infty$ | | | |
|---|---|---|---|---|---|---|---|---|
| | FMM | Lichtenstein et al. | MMP | Ours | FMM | Lichtenstein et al. | MMP | Ours |
| $x^2 - y^2$ | 0.02861 | 0.00213 | 0.00094 | **0.00043** | 0.0795 | 0.0117 | 0.0024 | **0.0016** |
| $x^2 + y^2$ | 0.02053 | 0.00339 | 0.00276 | **0.00096** | 0.0680 | 0.0144 | 0.0067 | **0.0026** |
| $x^2 - y^2 + xy$ | 0.02913 | 0.00336 | 0.00182 | **0.00088** | 0.0640 | 0.0260 | 0.0063 | **0.0028** |

### 4.3 GENERALIZATION TO ARBITRARY SHAPES

To better emphasize the generalization ability of our method, we conduct an additional experiment. We train our solver only on the three $2^{nd}$ order polynomial surfaces shown in Table 1, and evaluate it on arbitrary shapes from the TOSCA dataset (Bronstein et al., 2008). It can be seen in Figure 6, that our method generalizes well and leads to significantly lower errors compare to the heat method, classical fast marching and the method presented by Lichtenstein et al. (2019) when trained on the same polynomial surfaces. Errors are computed relative to the polyhedral distances, since they are the most accurate distances available to us for these shapes. For a more detailed analysis of the generalization of our method to Tosca, one can see Table 3 and Figure 9 in the Appendix.

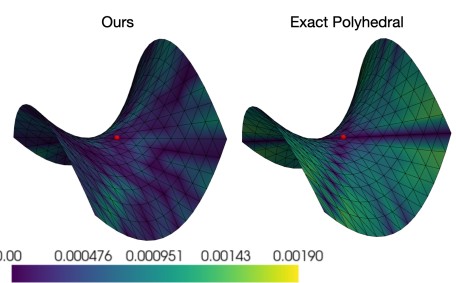

Figure 5: Polynomial surfaces: Errors presented for the polyhedral scheme and the proposed method. Local errors, represented as colors on the surface, were computed relative to exact polyhedral distances computed at a high-resolution sampled mesh of the continuous surface, as described in Section 3.3.

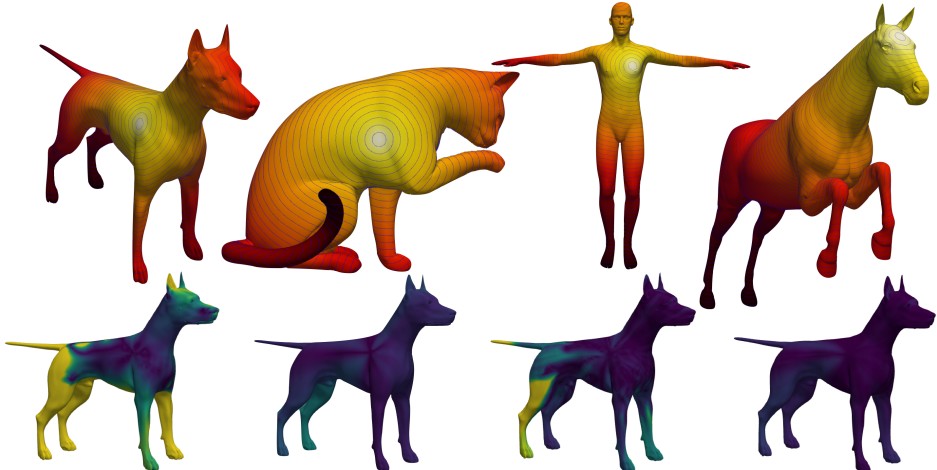

Figure 6: Generalization to arbitrary shapes. Top row: Iso-contours shown for our method. Bottom row: errors presented (left to right) for the heat method, fast marching, Lichtenstein et al. and the proposed method. Local errors presented as colors on the surface (brighter color indicates higher error), were computed relative to the polyhedral distances. The evaluation was conducted on TOSCA whereas our solver and the solver proposed by Lichtenstein et al. were trained with only limited number of $2^{nd}$ order polynomial surfaces (the 3 surfaces presented in Table 1).

### 4.4 ABLATION STUDY

To analyze the performance and robustness of our method, we conduct additional tests. These include modifying the local numerical support by which neighborhoods are defined and the precision point representation of the network weights.

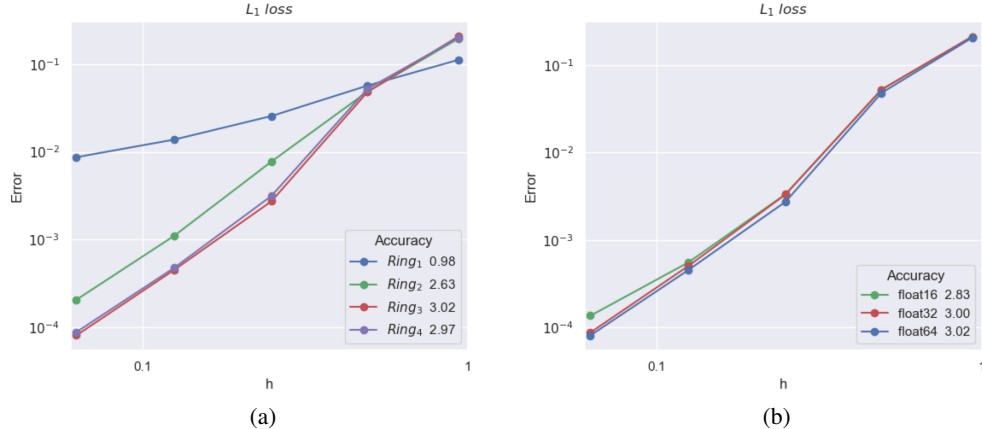

Figure 7: (a) Local neighborhood: Evaluation of the proposed method on spheres with different local neighborhoods support. $Ring_i$ corresponds to a neighborhood containing all vertices with at most $i$ edges from the evaluated target. (b) Precision floating point representation: Evaluation of our method on spheres with different precision floating point representation of the neural network weights.

**Local Neighborhood Support**. In the fast-marching method, the solver locally estimates a solution to the eikonal equation using a finite-difference approximation of the gradient. This approximation of the gradient is defined by a local stencil. For example, in the case of regularly sampled grids, the one-sided difference formula for a third order approximation requires a stencil with three neighboring points (Fornberg, 1988). In analogy to the stencil, our method uses a $3^{rd}$ ring neighborhood. Our local solver does not explicitly solve an eikonal equation nor does it use an approximation of the gradient. Yet, the size of the numerical support is the underlying ingredient that allows our neural network to realize high order accuracy. We have empirically validated this, as depicted in Figure 7a.

**Precision Floating Point Representation**. The choice of numerical representation is an important decision in the implementation of a neural network based solver. It leads to a trade-off between the accuracy of the solver and its execution time and memory footprint. In all our experiments, our main focus is on the accuracy of the method. Hence, we used double precision floating point for our neural network implementation. Figure 7b shows a comparison between our implemented network with different precision, showing only a slight degradation when a single and half precision floating points are used.

## 5 CONCLUSIONS

A fast and accurate method for computing geodesic distances on surfaces was presented. Inspired by Lichtenstein et al. (2019) we revisited the ingredients of dynamic programming based distance computation methodologies to designed a neural network based local solver. While the local solver proposed by Lichtenstein was limited to second order, we were able to lift this practical barrier by (1) extending the solver numerical support and (2) providing accurate high-order distances to train our solver for general surfaces for which there is no analytic expression for geodesic distances. We trained a neural network to locally extrapolate the values of the distance function on sampled surfaces. The result is a higher accuracy method compared to state of the art solvers while keeping the low quasi-linear computational complexity. To achieve a more accurate approximation of the distance function, the proposed solver is trained using an extended local numerical support (more neighboring points). For third order accuracy, the neighborhood consists of all vertices connected to the target point by a path with at most three edges. To train our solver using accurate examples, we proposed a novel multi-resolution generation bootstrapping technique that projects distances computed at high resolutions to lower ones. We believe that the proposed bootstrapping idea could be utilized for training other numerical solvers while keeping in mind that the numerical support enables the required accuracy.

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

## A APPENDIX

### A.1 EXACT DISTANCES ON A POLYHEDRON APPROXIMATING A CONTINUOUS SURFACE ARE SECOND ORDER ACCURATE

#### A.1.1 SECOND ORDER ACCURATE ON CIRCLES

When considering a polyhedral approximation of a continuous surface, the exact distances on the polyhedron would be $\mathcal{O}(h^2)$ approximation of distances on the continuous surface. Let us start with a toy example. Assume we try to approximate the circumference of a circle with radius $1$ in the plane using a regular polygon with $n$ vertices. Let $\theta = \frac{2\pi}{n}$ be the angle of the circular sector defined between two successive sample points on the circle. The distance between these points is given by $h = 2\sin(\frac{\theta}{2})$. The circumference of the circle is known to be $2\pi$, while the length approximated by the polygon is $nh = 2n\sin(\frac{\pi}{n})$. The truncation error would then be given by

$$
\begin{aligned}
2\pi - nh &= 2\pi - 2n\sin\left(\frac{\pi}{n}\right) \\
&= 2\pi - 2n\left(\frac{\pi}{n} - \frac{\left(\frac{\pi}{n}\right)^3}{3!} + \frac{\left(\frac{\pi}{n}\right)^5}{5!} - \cdots\right) \\
&= \frac{\pi^3}{3n^2} - \frac{\pi^5}{60n^4} + \cdots \\
&= \mathcal{O}\left(n^{-2}\right) \\
&= \mathcal{O}(h^2).
\end{aligned}
\tag{7}
$$

Note, that this analysis also provides a lower bound on the length estimation error of great circles on a sphere. In order to analyze the behaviour of more general surfaces approximated by triangulated polyhedral surfaces we will resort to the general proof in A.1.2

#### A.1.2 SECOND ORDER ACCURACY FOR GENERAL SURFACES

Let $S : \Omega \in \mathbb{R}^2 \to \mathbb{R}^3$ be a Riemannian two dimensional manifold with effective Gaussian curvature a.e. Let $C(s) : [0, L] \to S$ be a minimal geodesic connecting two surface points $C(0)$ and $C(L)$ on $S$ with arclength parametrization $s$, and $L$ the length of $C$. We prove that the length of $C$ differs by $\mathcal{O}(h^2)$ from the sum of the lengths of the cords. These line segments, of length $h$ each as measured in $\mathbb{R}^3$, are defined by a sequence of surface points $C(s_i)$ and $C(s_{i+1})$. That is, the length of the approximation $\gamma$ defined by its vertices $\{C(0), C(s_2), \ldots, C(s_{n-1}), C(L)\}$, is given by

$$
L(\gamma) = \sum_{i=1}^{n-1} \|C(s_{i+1}) - C(s_i)\|_{\mathbb{R}^3} = nh,
\tag{8}
$$

differs by $\mathcal{O}(h^2)$ from

$$
L(C) = \int_0^L ds.
\tag{9}
$$

Consider the length parameterization along the line segment with end points $C(s_i)$ and $C(s_{i+1})$ be given by $t \in [-h/2, h/2]$, and assume w.l.o.g. the monotone increasing reparametrization $s(t)$ that would allow us to parametrize the surface geodesic segment between $C(s_i)$ and $C(s_{i+1})$. As $t$ is the arclength along the cord connecting the two end points of the line segment, by freedom of parametrization, we could choose $|C_t(0)| = 1$.

Next, lets compute the length of $C(t)$ in the $i$th interval.

$$
\begin{aligned}
L_S(C(s_i), C(s_{i+1})) &= \int_{s_i}^{s_{i+1}} ds \\
&= \int_{-h/2}^{h/2} |C_t| dt.
\end{aligned}
\tag{10}
$$

Lets expand $|C_t|$ about $0$, by which we have

$$
|C_t(t)| = |C_t(0)| + t\left(\frac{d}{dt}|C_t|\right)(0) + \frac{t^2}{2}\left(\frac{d^2}{dt^2}|C_t|\right)(0) + \cdots
$$

$$= |C_t(0)| + t\left(\frac{\langle C_t, C_{tt}\rangle}{|C_t|}\right)(0) + \frac{t^2}{2}\left(\frac{d}{dt}\frac{\langle C_t, C_{tt}\rangle}{|C_t|}\right)(0) + \cdots \qquad (11)$$

Let us focus on the second term,

$$\frac{\langle C_t, C_{tt}\rangle}{|C_t|} = \frac{\langle C_t, C_{tt}\rangle}{|C_t|^3}|C_t|^2 = \kappa|C_t|^2, \qquad (12)$$

where $\kappa$ is the curvature (normal curvature for a geodesic) of $C$ at that point. The third term is given by

$$\begin{aligned}
\frac{d}{dt}\kappa|C_t|^2 &= \kappa_t|C_t|^2 + 2\kappa\langle C_t, C_{tt}\rangle \\
&= \kappa_s|C_t|^3 + 2\kappa^2|C_t|^3. 
\end{aligned} \qquad (13)$$

We conclude with

$$\begin{aligned}
L_i &= \int_{s_i}^{s_{i+1}} ds \\
&= \int_{-h/2}^{h/2} |C_t| dt \\
&= \int_{-h/2}^{h/2}\left(|C_t(0)| + t\left(\kappa|C_t|^2\right)(0) + \frac{t^2}{2}\left(\kappa_s|C_t|^3 + 2\kappa^2|C_t|^3\right)(0) + \cdots\right) dt \\
&= |C_t(0)|h + 0 + \left(\kappa_s|C_t|^3 + 2\kappa^2|C_t|^3\right)(0)\frac{t^3}{6}\Big|_{-h/2}^{h/2} + \mathcal{O}(h^5) \\
&= |C_t(0)|h + \left(\kappa_s|C_t|^3 + 2\kappa^2|C_t|^3\right)(0)\frac{h^3}{24} + \mathcal{O}(h^5). 
\end{aligned} \qquad (14)$$

With our specific selection of $|C_t(0)| = 1$, we conclude with the overall error given by

$$\begin{aligned}
\mathcal{E}rr &= \sum_{i=1}^{n-1}|L_i - h| \\
&= \sum_{i=1}^{n-1}\left|(\kappa_s + 2\kappa^2)\frac{h^3}{24} + \mathcal{O}(h^5)\right| \\
&= h^3\sum_{i=1}^{n-1}\left|\left(\frac{1}{24}\kappa_s + \frac{1}{12}\kappa^2\right) + \mathcal{O}(h^2)\right| \\
&= \mathcal{O}(h^3)\mathcal{O}(n) 
\end{aligned} \qquad (15)$$

where $\kappa$ and $\kappa_s$ are evaluated at $t = 0$ for each segment. Note, that $\kappa$ and $\kappa_s$ are geometric quantities and thus could be regarded as effective bounded constants. Then, by the assumption $h \approx \mathcal{O}(n^{-1})$ we proved the convergence rate to be $\mathcal{O}(h^2)$.

## A.2 NEURAL ARCHITECTURE ENGINEERING CONSIDERATIONS

In this research we aim to develop high order accurate methods for computing geodesic distances on surfaces. To overcome the 2nd order restriction of the polyhedral approximation, we start with the method presented in Lichtenstein et al. (2019). This method uses a neural network based local solver that circumvents the polyhedral representation of the surface and operates directly on the neighboring points. In the fast-marching method, the local solver estimates a solution to the eikonal equation using a finite-difference approximation of the gradient. The stencil size in the finite difference method determines the accuracy of the operator (Fornberg, 1988). Our hypotheses is that extending the numerical support of the local solver would improve the overall order of accuracy.

In our first attempt we examine the neural network presented by Lichtenstein et al. (2019) while extending the local support to 3rd ring neighborhood. Such a direct extension did not improve the accuracy of the method, as can be seen in Figure 8. Next, we studied the trained model latent vector obtained after the max-pooling operation. Out of the 1024 entries of this vector, only 96 were different than zero.

Based on this observation we experimented with architectural modifications by extending the number of hidden layers, changing the activation functions, and reducing the size of the latent space. These modifications are presented in Table 2, where Model 1 is our proposed neural architecture presented in Figure 2, and all other models have a similar architecture up to the changes defined in the table. As can be seen in Figure 8, the various architectural changes have a large impact on the performance of the solver .

Table 2: Neural Architectures: Various architectural modifications to our presented neural network 2. The residual connections and the different activation functions are defined only for the shared weight MLP, while the fully connected network applied after the pooling operation has LeakyRelu activations with a negative slope of 0.001 in all variants of our neural network.

| Neural architecture | Residual connections | Activation function | negative slope | Pooling operation |
|---|---|---|---|---|
| Model 1 | ✓ | LeakyReLu | 0.2 | max |
| Model 2 | ✓ | LeakyReLu | 0.2 | avg |
| Model 3 | ✓ | LeakyReLu | 0.001(default) | max |
| Model 4 | ✓ | ReLu | ✗ | max |
| Model 5 | ✗ | ReLu | ✗ | max |
| Lichtenstein et al. | ✗ | ReLu | ✗ | max |

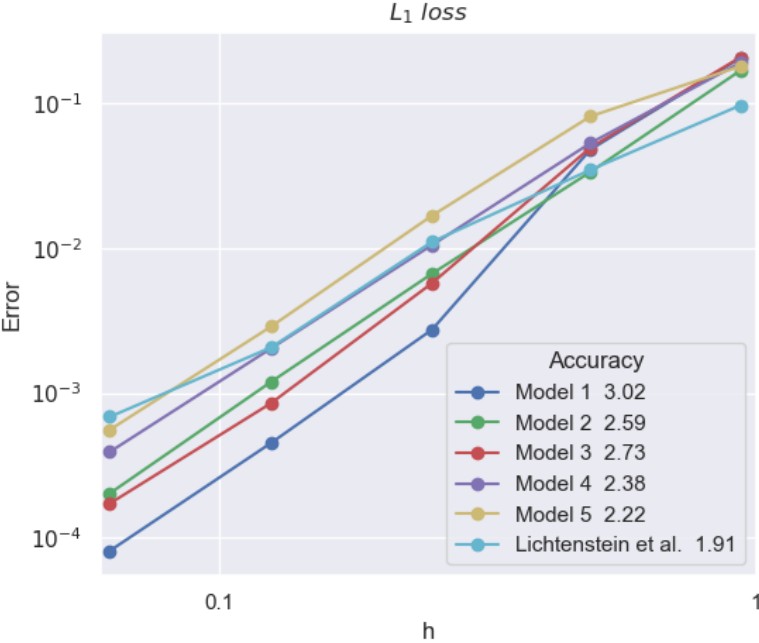

Figure 8: Neural architectures: Evaluation of the effects of different modifications to our neural network, presented in 2. All different models use the same local neighborhood support, which includes all vertices with at most three edges from the evaluated target.

## A.3 ROBUSTNESS TO TRIANGULATION

The proposed local solver is applied directly to the mesh points, and does not use the underlying triangulation. However, the numerical support of the solver and the distance updating scheme do depend on the triangulation. Therefore, it is important to investigate the robustness of our method to various triangulation methods. Ones that was not part of the training of the proposed solver.

Table 3: Generalization to arbitrary shapes: Quantitative evaluation tested on TOSCA. The errors are relative to the polyhedral scheme. Our solver and the solver proposed by Lichtenstein et al. were trained using our bootstrapping method 3.3 with a very limited number of $2^{nd}$ order polynomial surfaces (the 3 surfaces presented in Table 1).

| Shape | $L_1$ | | | | $L_\infty$ | | | |
|---|---|---|---|---|---|---|---|---|
| | Heat | FMM | Lichtenstein et al. | Ours | Heat | FMM | Lichtenstein et al. | Ours |
| Dog | 0.0728 | 0.0110 | 0.0123 | **0.0037** | 0.8688 | 0.1514 | 0.1318 | **0.0465** |
| Cat | 0.1596 | 0.0136 | 0.0386 | **0.0053** | 0.6541 | 0.0631 | 0.3450 | **0.0384** |
| Wolf | 0.0440 | 0.0162 | 0.0169 | **0.0072** | 0.2244 | 0.1009 | 0.1343 | **0.0422** |
| Horse | 0.1084 | 0.0136 | 0.0199 | **0.0068** | 0.8239 | 0.2273 | 0.1616 | **0.0896** |
| Michael | 0.1185 | 0.0109 | 0.1073 | **0.0054** | 0.5906 | 0.1602 | 0.4873 | **0.0881** |
| Victoria | 0.1211 | 0.0076 | 0.0479 | **0.0027** | 0.4538 | 0.1075 | 0.2289 | **0.0414** |
| Centaur | 0.0672 | 0.0188 | 0.1088 | **0.0062** | 4.9437 | 0.1985 | 0.7348 | **0.1417** |

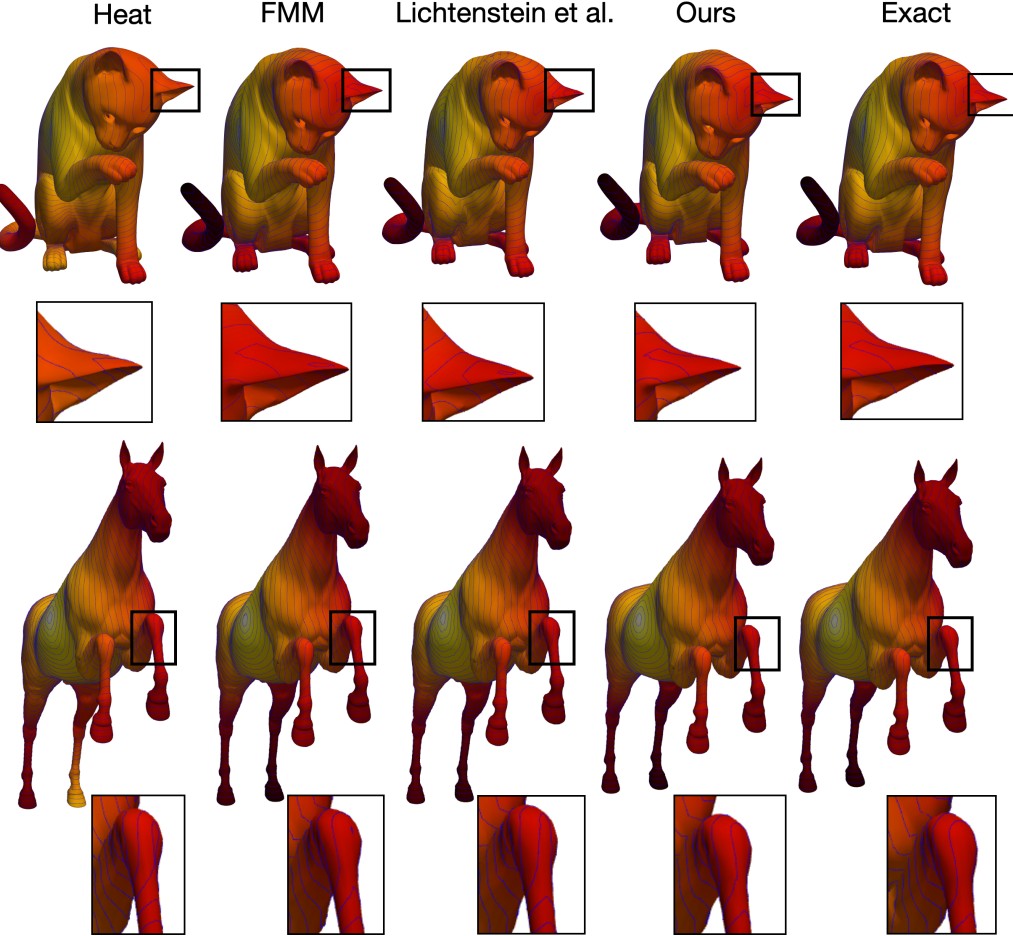

Figure 9: Generalization to arbitrary shapes: Iso-contours shown for (left to right) the heat method, fast marching, Lichtenstein et al., our method and the exact polyhedral scehme, calculated by the MMP algorithm. The evaluation was conducted on shapes from TOSCA whereas our solver and the solver proposed by Lichtenstein et al. were trained with only limited number of $2^{nd}$ order polynomial surfaces (the 3 surfaces presented in Table 1

In a new experiment we train our neural network on regularly sampled spheres and evaluate the resulting solver on arbitrarily triangulated spheres that includes ill-posed triangles. We show the numerical and qualitative evaluation in Table 4 and Figure 10. It can be seen that the proposed method is robust to different triangulations and produces lower errors compared to the heat method,

the classical fast marching, the exact geodesic method, and the deep learning method of Lichtenstein et al.

Table 4: Robustness to triangulation: Quantitative evaluation of the ability of the proposed method to handle triangulations other than those for which it was trained for. The evaluation was conducted on randomly uniformly non-regularly sampled spheres whereas our solver and the solver proposed by Lichtenstein et al. were trained only on regularly sampled spheres.

| Error | FMM | Heat method | Lichtenstein et al. | MMP | Ours |
|---|---|---|---|---|---|
| $L_1$ | 0.01804 | 0.01964 | 0.00510 | 0.00128 | **0.00099** |
| $L_2$ | 0.01865 | 0.02048 | 0.00672 | 0.00142 | **0.00127** |
| $L_\infty$ | 0.0270 | 0.0361 | 0.02278 | **0.0027** | 0.0041 |

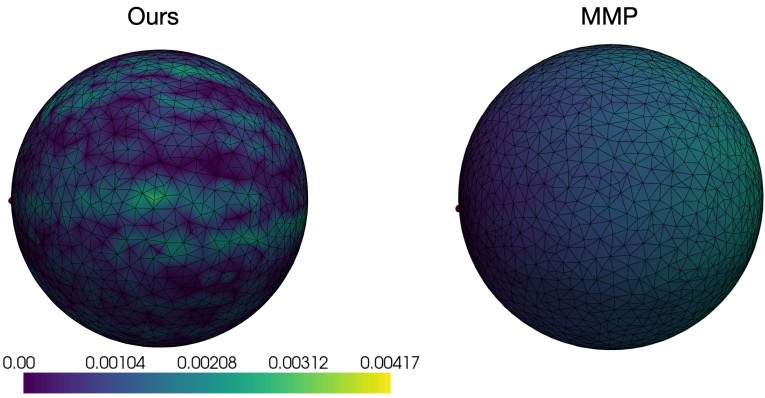

Figure 10: Robustness to Triangulation: Errors presented for the "exact" MMP scheme and the proposed method. Local errors, represented as colors on the surface, were computed relative to the analytical geodesic distances. The proposed method was trained on regularly sampled spheres as can be seen in Figure 12.

## A.4 THE UPDATE STEP COMPLEXITY

Theoretically, our method has the same quasi-linear computational complexity as the fast marching method and the method presented by Lichtenstein et al. (2019), all of which relies on the distance updating scheme presented in Algorithm 1. The different local solvers, while having $\mathcal{O}(1)$ computation complexity, involve different constants.

A single update step of the proposed method requires $1537\mu s$, a single update step in Lichtenstein et al. (2019) requires $788\mu s$, while an update in the fast marching method requires only $6.3\mu s$. Hence, the fast marching is approximately 250 times faster when applied to the same grid resolution. And yet, the high order accuracy allows us to sub-sample our triangulated surfaces and save on overall complexity achieving the same goals.

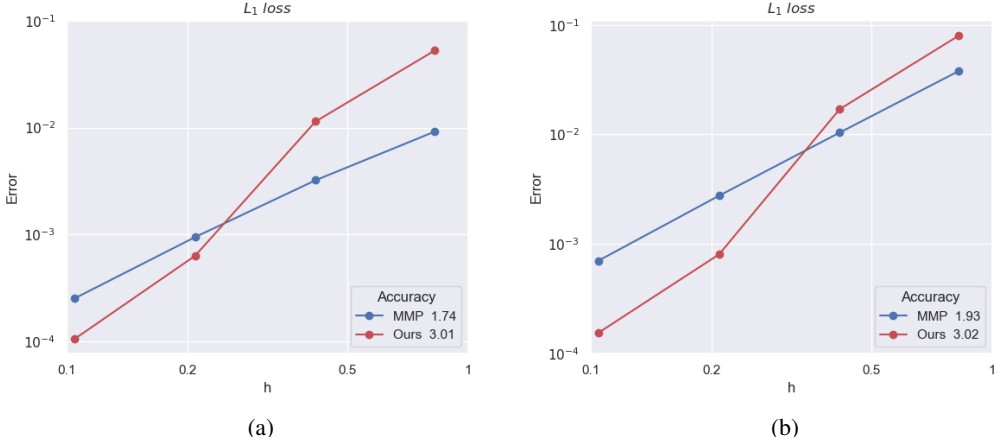

(a)                                                     (b)

Figure 11: Order of accuracy on parametric surfaces: Plots showing the effect of the edge size on the errors. The accuracy of each scheme is associated with its corresponding slope. Ground truth distances were evaluated using our bootstrapping method 3.3. (a) Evaluation on hyperbolic paraboloid $x^2 - y^2$. (b) Evaluation on regular paraboloid $x^2 + y^2$.

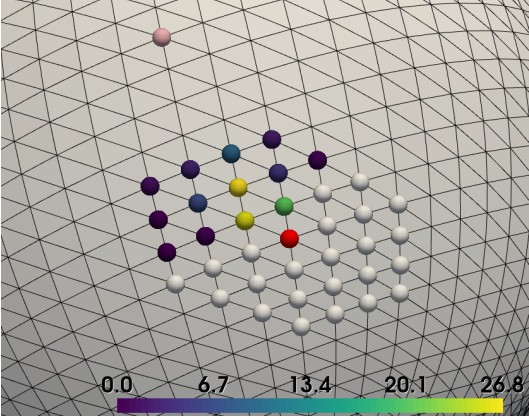

Figure 12: Attention map showing the importance of each neighboring point in the network approximation. The source point is colored pink, the point whose distance is currently evaluated is colored red, and all other plotted points form the current local neighborhood. The *Visited* neighboring points that are fed into the neural network are colored according to the percentage of the corresponding features in the latent vector obtained from the max-pooling operation.

