# OpenReview forum: "DEEP ACCURATE SOLVER FOR THE GEODESIC PROBLEM"
_ICLR.cc/2023/Conference — Submitted to ICLR 2023_

### Official Review · Reviewer_58bg · 2022-10-20

**Confidence:** 4
**Clarity, Quality, Novelty And Reproducibility:** see the strength section.
**Correctness:** 4
**Technical Novelty And Significance:** 3
**Empirical Novelty And Significance:** 2
**Recommendation:** 3

**Strength And Weaknesses:**

Strengths:
1. The method is simple. Can generalise to complex shapes easily.
2. The paper is well-written. The proposed concepts are well-explained. There are enough details in the paper to reproduce the method.
3. The results show a huge improvement over sota.

Weakness:
1. The technical novelty is limited to using pointnet features to learn geodesic distances.
2. My main concern is the experiment section. The complexity of the method is mentioned but the concrete timing performance of the method is never talked about. I understand that the distance update steps is going to expensive but it is important to see the difference.
3. The quantitative evaluation is missing on TOSCA dataset. The authors should have reported the geodesic computation accuracy on this dataset and should have compared with other methods such as FMM, MMP and Lichtenstein et al. Perhaps, MMP can be used as baseline.
4. Since the method is trained on spheres of arbitrary radii (assuming that they vary between extremely small and large), it would have been nice to see the localised performance comparison on the sharp regions such as cat ears, dog-tails etc.
5. FMM is missing from table 1.

**Summary Of The Paper:**

This paper presents a learning-based method for computing geodesics on a triangulated mesh. The algorithm is similar to fast marching cubes with O(NlogN) complexity, except the distance updating step which is the one of the two main contributions of the paper.
The proposed distance update step encodes the 3-ring neighbors along with distances from the source (normalised such that norm of distances is 1) using Pointnet (existing method for encoding 3d data) to get a representation of each neighborhood. Then the method tries to train this encoder on ground truth data in a supervised fashion.
The second main contribution is that, since the method is computing local geodesics, it is trained on spherical surfaces (simple analytical surfaces)  where an exact computation of geodesics is possible. The training is completely supervised. It is very easy to see that once trained on this data, the method can be easily extended to any complex shape.
The experiments demonstrate superior geodesic computation wrt main sota methods.

**Summary Of The Review:**

I think that the technical novelty is limited and the lack of experimental evaluation does not allow me to overlook that part. Had it been a (learning-based) method X times faster and X times more accurate (accuracy has been demonstrated partially on polynomial surfaces) on real data (TOSCA, but some other datasets. should have been used as well), my opinion could have been different as it could have been a method with limited novelty but significantly improved performance.

---

> ### Author Response · Authors · 2022-11-08
> **The proposed method is more exact than the so called ``exact'', and the bootstrapping allows to push it forward. Please see details and re-evaluate.**
>
> We appreciate your efforts in evaluating our paper.
> As you stated the method "Can generalise to complex shapes easily" and ``The results show a huge improvement over sota".
> Please see our responses to the raised concerns.
> We will be happy to address further unclear issues upon request.
>
> >``The technical novelty is limited to using pointnet features to learn geodesic distances"
>
> The main novelty can be found in two main aspects.
> 1. It is shown that by extending the numerical support of the NN one can lift the 2nd order accuracy barrier for approximating geodesic distances.
> 2. Introduction of the bootstrapping idea that allows generating accurate distances beyond 2nd order for surfaces.
> It allowed us to train the NN and evaluate the performance of our and other solvers on surfaces beyond spheres.
>
> As now shown in the revised version, a simple extension of Lichtenstein's et al. 2nd neighbors to 3rd ring neighborhood does not provide a 3rd order accurate scheme.
> The proposed neural network was chosen after analyzing the NN proposed by Lichtenstein et al.
> Lichtenstein et al. architecture suffers from data propagation issues and a low dimensional latent space. Our analysis and engineering considerations are added to the revised version.
> Indeed, Figure 7(a) shows that the proposed network architecture with 2nd ring neighborhood significantly outperforms the network introduced by Lichtenstein et al.
>
> >``My main concern is the experiment section. The complexity of the method is mentioned but the concrete timing performance of the method is never talked about"
>
> The update step is efficient - timings are added.
>
> >``The quantitative evaluation is missing on TOSCA dataset"
>
> There is a fundamental point that we have failed to communicate: The proposed method is more accurate than the so called "exact" one.
> So, there is no "ground truth" to compare with when considering the TOSCA data set.
> This lack of ground truth data for general surfaces gave birth to our bootstrapping idea which we consider innovative.
>
> >``Since the method is trained on spheres of arbitrary radii (assuming that they vary between extremely small and large), it would have been nice to see the localised performance comparison on the sharp regions such as cat ears, dog-tails etc."
>
> It is important to state that the method which was evaluated on shapes from TOSCA was not trained on spheres, but on polynomial surfaces using the bootstrapping idea.
> Indeed, local regions can be well approximated by polynomials, while the order of such polynomials would determine the rate of convergence.
>
> >``FMM is missing from table 1"
>
> Added in the revised version, though see above comments.

---

### Official Review · Reviewer_ssEx · 2022-10-25

**Confidence:** 4
**Clarity, Quality, Novelty And Reproducibility:** I found the quality of writing to be …
**Correctness:** 4
**Technical Novelty And Significance:** 3
**Empirical Novelty And Significance:** 3
**Recommendation:** 8

**Strength And Weaknesses:**

Strengths:
- Well-written and clear
- The approach clearly improves upon existing methods, including one with a related approach (Liechtenstein et al.).
- A simple and sensible neural network architecture is used.

Weaknesses:
- As Liechtenstein et al. considered a very similar idea already, it could be argued that the method does not have sufficient novelty.
- The work does not explore the use of other neural network architectures for the local solver, or at least does not comment on it.
- It is unclear if the experiments show robustness to the quality of the triangulations for the underlying meshes. Certainly, it seems that the training set triangulations are well-conditioned, so it'd be curious to see its behavior on poorly conditioned meshes.

**Summary Of The Paper:**

The paper formulates a third-order accurate fast-marching style algorithm for finding geodesic distances on surfaces embedded in 3D. The local solver is a trained neural network which operates on the 3-ring of the point in question. The algorithm has quasilinear complexity ultimately. For training, a multi-resolution approach is used, resulting in a method that can find distances on coarse meshes that are more accurate than would be possible with exact methods.

The method is compared to an exact solver, MMP, and a related work Liechtenstein et al. which looks at the 2-ring and forms a second-order accurate scheme. The comparisons are favorable. Ablation studies show that increasing the neighborhood support further does not improve the accuracy of the method.

**Summary Of The Review:**

The strengths and weaknesses above summarize my main thoughts on the work. I feel that the work is well-written and investigated, with clear motivation and rationale behind the method. Moreover, it improves upon existing methods, so I am for acceptance.

---

> ### Author Response · Authors · 2022-11-08
> **Thank you for your constructive review and positive feedback.**
>
> Thank you for your constructive review and positive feedback.
> As for comparison to Lichtenstein et al., if we ignore the complexity, their accuracy is the same as for the MMP ``exact" algorithm, while the proposed method is the first to lift the 2nd order barrier, and provide a practical method for generating accurate distances for general surfaces.
>
> >``The work does not explore the use of other neural network architectures for the local solver, or at least does not comment on it.''
>
> Excellent point.
> The proposed neural network was chosen after analyzing the NN proposed by Lichtenstein et al.
> Their architecture suffered from data propagation issues and a low dimensional latent space.
> Our analysis and engineering considerations is added to the revised version.
>
> >``It is unclear if the experiments show robustness to the quality of the triangulations for the underlying meshes"
>
> We provide more experiments that demonstrate the robustness of the method to ill-conditioned triangulated meshes.

---

### Official Review · Reviewer_UqX8 · 2022-10-26

**Confidence:** 4
**Correctness:** 2
**Technical Novelty And Significance:** 2
**Empirical Novelty And Significance:** 2
**Recommendation:** 3

**Clarity, Quality, Novelty And Reproducibility:**

## Clarity
The paper is mostly clear.
- The first paragraph of section 3.1 could be reworked a bit. You have "It can replace the distance approximation method...," but no other method is presented to be replaced. "The neighboring points ... are defined by the distance of the connectivity graph" is a bit confusing, suggesting that the neighborhood depends on the distances being computed. It should be enough to say that the neighborhood includes all vertices connected to $p$ by a path of at most 3 edges, and leave out the rest of the paragraph.
- In section 3.2 and elsewhere, you refer to "the causal nature of our algorithm," but what you mean by "causal" is never explained. I am assuming you mean the local solver should only consider points that would have been previously visited by the Dijkstra-like algorithm. Please state this more clearly or don't use the word "causal."

## Quality and Novelty
The only differences from Liechtenstein et al. appear to be (1) the size of local neighborhood considered; and (2) the multiresolution bootstrapping scheme for training. While these are not huge innovations, they seem to lead to significantly better accuracy. However, more extensive evalutaion is necessary.

**Strength And Weaknesses:**

## Strengths
- The method is presented fairly clearly.
- The few results that are presented look good—namely Figure 4 showing empirical distance accuracy of order $h^3$ on meshes of a sphere, and Figure 6 showing better accuracy than competing methods on a dog mesh.
## Weaknesses
- The method is claimed to be 3rd-order accurate, i.e., $\mathcal{O}(h^2)$ throughout the paper. However, this appears to be only an empirical statement based on evaluation on spheres. If you want to claim this accuracy, more evaluation is necessary, especially on real data surfaces from TOSCA. The evaluation of generalization from polynomial surfaces to TOSCA surfaces is very quick and only visual. A more careful and comprehensive numerical evaluation of generalization to the TOSCA dataset is necessary to make a convincing case that this method is highly accurate.
- The paper claims that "exact geodesic distances computed on a polygonal mesh approximating a continuous surface would be at most a second order approximation," and Appendix A provides an argument to that effect. However, the argument is restricted to a very special case (polygonal approximation of circles in the plane), and it is not a proof. Moreover, no proof is cited. To make a strong claim like this, proof or citation should be given.
- Section 3.3 (bottom of page 6) claims that the bootstrapping approach can achieve arbitrary accuracy. This seems like it would only be true if the generalization error of your learned local solver is zero. Please provide further justification or else remove these claims.

**Summary Of The Paper:**

This paper extends the Deep Eikonal Solver of Liechtenstein et al. It achieves higher empirical accuracy by allowing the local solver to see a 3-ring rather than 2-ring neighborhood, and by training against ground truth distances computed on a finer mesh. The results display generalization from polynomial surfaces to surfaces from the TOSCA dataset.

**Summary Of The Review:**

The paper shows some very accurate results in limited evaluations, building on the work of Liechtenstein et al. However, it makes some very strong and unsubstantiated claims that would require more extensive empirical evaluation of generalization performance to justify.

---

> ### Author Response · Authors · 2022-11-08
> **There is a fundamental point that was most probably missed.**
>
> Thank you for finding the empirical results to be strong, and the paper write-up to be clear.
> We will try to answer all your concerns and be happy to address additional questions during the discussion period.
>
> >``The method is claimed to be 3rd-order accurate, i.e. ${\cal O}(h^3)$, throughout the paper. However, this appears to be only an empirical statement based on evaluation on spheres. If you want to claim this accuracy, more evaluation is necessary, especially on real data surfaces from TOSCA"
>
> Geodesic distances can be computed analytically only for planes and spheres.
> It is thus common to evaluate the order of accuracy of numerical schemes on such surfaces.
> And yet, our novel bootstrapping method allows us to evaluate the order of accuracy for more general surfaces.
>
> All TOSCA surfaces are synthetic, and as shown in the paper, the most accurate known method for computing distances on surfaces are the so-called ``exact''-MMP or Lichtenstein's  algorithms which are ${\cal O}(h^2)$ accurate.
>
> >``The paper claims that "exact geodesic distances computed on a polygonal mesh approximating a continuous surface would be at most a second order approximation," and Appendix A provides an argument to that effect. However, the argument is restricted to a very special case"
>
> Integrating any function along a path embedded in a polyhedral surface approximating a continuous one is at most 2nd order according to the trapezoidal rule error formula.
> It applies to any smooth surface, and in fact, any  function integrated along such discretizations rather than just the length.
> If one takes the toy proof provided in the Appendix as a motivation and consider geodesics on spheres $S^2$ in $\mathbb{R}^3$ rather than a circle $S^1$ in the plane $\mathbb{R}^2$, the current proof still applies.
>
> >"The evaluation of generalization from polynomial surfaces to TOSCA surfaces is very quick and only visual. A more careful and comprehensive numerical evaluation of generalization to the TOSCA dataset is necessary to make a convincing case that this method is highly accurate."
>
> There is a fundamental point that was missed altogether. The proposed method is more accurate than the “exact” method, so there is no “ground truth” reference to compare with when considering the TOSCA data set.
> And yet, we added a comparison on TOSCA comparing to the MMP 2nd order scheme.
>
> >``Section 3.3 (bottom of page 6) claims that the bootstrapping approach can achieve arbitrary accuracy. This seems like it would only be true if the generalization error of your learned local solver is zero"
>
> “Generalization” here is nothing but the universality of scaling and sampling.
> If one accepts the fact that there is a given ${\cal O}(h)$ accurate solver, then, for a sub-sampled surface with $\tilde{h} = h^2$, solving for that $\tilde{h}$ resolution with the ${\cal O}(\tilde{h})$ solver, one obtains ${\cal O}(h^2)$ to train by.
> This is an important point that we feel was missed.
> It has little to do with the notion of model generalization.
>
> As for the contribution.
> Note that Lichtenstein et al. proposed an ${\cal O}(h^2)$ at quasi-linear time.
> If we ignore the complexity, the accuracy is the same as the MMP-“exact” algorithm.
> Here, the proposed linear method is more exact than the MMP-``exact" quadratic algorithm.
> If one ignores the technicalities for a moment, this should be appreciated for its own right as an empirical novelty and significance.
>
> As for architecture, the local solver we propose was refined to allow better data propagation.
> This allows us to benefit from the extended support that was impossible in Lichtenstein et al. network architecture.
> We added an analysis section regarding such engineering considerations for the network architecture in the revised version.
> Figure 7(a) shows that the proposed network architecture with 2nd ring neighborhood significantly outperforms the network introduced by Lichtenstein et al.

---

### Author Response · Authors · 2022-11-17
**Rebuttal paper revision summary**

We would like to thank the reviewers for their valuable feedback that helped us improve the presentation of the method and the results.
We appreciate your efforts in evaluating our submission.

In this revised version (18/11/2022) we have:

* Included a bound on the rate of convergence of computing distances on polyhedral approximations of  general surfaces.
Indeed, exact geodesic distances computed on a polygonal mesh approximating a continuous surface would be at most a second order accurate (Appendix A.1.2, as suggested by Reviewer UqX8).
* Included a detailed quantitative evaluation on the Tosca dataset, showing a strong generalization from simple 2nd order polynomial surfaces to arbitrary surfaces (Table 3, as suggested by reviewers UqX8 and 58bg).
* Included localized performance comparison on sharp regions such as cat ears and horse knees (Figure 9, as suggested by Reviewer 58bg).
* Included an experiment showing robustness to triangulation (Appendix A.3, as suggested by Reviewer ssEx).
* Added the performance evaluation of the fast marching method to the comparison on parametric surfaces (Table 1, as suggested by Reviewer 58bg).
* Included neural architecture engineering considerations and comparisons of the different modifications to the neural network.
  We show that just extending the numerical support of the architecture presented by Lichtenstein et al. is insufficient for reaching 3rd order of accuracy (Appendix A.2, as discussed by Reviewes UqX8 and 58bg).
* Included concrete timing performance of the update step compared to other methods (Appendix A.4, as suggested by Reviewer 58bg).
* Included an evaluation of the order of accuracy on parametric surfaces, while using our bootstrapping method to generate accurate ground truth distances. Indeed, our method is shown to be 3rd order accurate on surfaces other than spheres  (Figure 11, as discussed with Reviewer UqX8).

To the best of our understanding all the questions and concerns raised by the reviewers were addressed and answered.
We hope the reviewers would find the modified version of our submission  worthy for publication in ICLR and consider upgrading their previous scores.
If there are any other questions or concerns, we would be happy to continue the discussion.

---

### Decision · Program_Chairs · 2023-01-20

**Decision:**

Reject

**Justification For Why Not Higher Score:**

N/A

**Justification For Why Not Lower Score:**

N/A

**Metareview: Summary, Strengths And Weaknesses:**

Thank you for submitting your work to ICLR 2023. The main issues with this work seems to be: 1) its contribution is incremental compared to that of Lichtenstein et al. 2019, which already provides the main methodology used in this paper; and 2) any claims of accuracy of the method must be supported by experiments on a suitably rich and broadly representative test set.
During discussions among reviewers the reviewer with the high score was unwilling to champion the paper and agreed with the main limitations detailed above.